# Identification of Human Breast Adipose Tissue Progenitors Displaying Distinct Differentiation Potentials and Interactions with Cancer Cells

**DOI:** 10.3390/biomedicines10081928

**Published:** 2022-08-09

**Authors:** Pascal Peraldi, Agnès Loubat, Bérengère Chignon-Sicard, Christian Dani, Annie Ladoux

**Affiliations:** 1CNRS, INSERM, iBV, Université Côte d’Azur, 06107 Nice, France; 2Department of Plastic and Reconstructive Surgery, Pasteur 2 Hospital, Université Côte d’Azur, 06107 Nice, France; 3CNRS, Institute of Biology Valrose (iBV), University of Nice Sophia-Antipolis, 28 Avenue de Valombrose, CEDEX 2, 06107 Nice, France

**Keywords:** breast adipose progenitors, breast cancer, adipocytes, primary cilium, myofibroblasts, CD274, TGFβ, TNFα, ER receptors, tumor invasion

## Abstract

Breast adipose tissue (AT) participates in the physiological evolution and remodeling of the mammary gland due to its high plasticity. It is also a favorable microenvironment for breast cancer progression. However, information on the properties of human breast adipose progenitor cells (APCs) involved in breast physiology or pathology is scant. We performed differential enzymatic dissociation of human breast AT lobules. We isolated and characterized two populations of APCs. Here we report that these distinct breast APC populations selectively expressed markers suitable for characterization. The population preferentially expressing *ALPL* (MSCA1) showed higher adipogenic potential. The population expressing higher levels of *INHBA* and *CD142* acquired myofibroblast characteristics upon TGF-β treatment and a myo-cancer-associated fibroblast profile in the presence of breast cancer cells. This population expressed the immune checkpoint CD274 (PD-L1) and facilitated the expansion of breast cancer mammospheres compared with the adipogenic population. Indeed, the breast, as with other fat depots, contains distinct types of APCs with differences in their ability to specialize. This indicates that they were differentially involved in breast remodeling. Their interactions with breast cancer cells revealed differences in the potential for tumor dissemination and estrogen receptor expression, and these differences might be relevant to improve therapies targeting the tumor microenvironment.

## 1. Introduction

Adipose tissue (AT) is disseminated throughout the body, and its primary function is to store excess calories as lipids to maintain energy hemostasis and metabolic health. Lipids are stored in droplets within adipocytes, and this helps prevent the harmful effects of ectopic fat accumulation in the organs. Adipocytes are the main cell type of the AT. Upon metabolic demand, new adipocytes are produced from adipose progenitor cells (APCs), which are also present in the adipose tissue. For many years, APCs were identified as fibroblasts expressing Pref-1 (DLK-1) [1]. However, the diversity of adipose depots; their distinct embryonic origins, i.e., mesodermal or neural crest derived [2]; and the lobular architecture of adipose depots composed of a septa and stroma have contributed to the isolation of heterogeneous APC populations. Therefore, their contribution to the remodeling of adipose depots, either under physiological or pathological conditions, remains an open question.

There are different types of adipose tissue in adults [3]. White adipose tissue (WAT) stores energy and controls body weight [4]. In contrast, brown adipose tissue (BAT) dissipates energy to promote non-shivering thermogenesis [5]. In addition, brown or beige adipocytes, which are derived from white adipocytes while exhibiting the properties of brown adipocytes, have been identified [6,7]. Although the adipose tissue of the breast is predominantly WAT, a fourth type of adipocytes has been described in the breast during pregnancy and lactation: pink adipocytes [8]. They are involved in milk production and disappear when lactation stops. Regardless of the AT type, adipocytes are normally never in contact with other cells of an organ, except during involution of the mammary gland after weaning, where they are close to the epithelial cells [9]. In healthy adults, adipocytes are separated from epithelial cells by a basement membrane, thus preventing their interactions. The lobular architecture of the adipose tissue is a key determinant to delineate the fate of progenitor cells [10]. Indeed, lobules are composed of distinct extracellular matrix components that have been described for many adipose tissue depots: septa and stroma that define niches for APCs [11]. Depending on their location, distinct types of APCs have been identified by the expression of specific markers [12]. In general, CD45-/CD34+/CD31− progenitor cells are enriched in the stroma and tend to differentiate into adipocytes when they express ALPL (or MSCA1). In contrast, those with low levels of MSCA1 but high levels of INHBA are preferentially found in the septa and tend to specialize into myofibroblasts [11].

While many studies have focused on subcutaneous and visceral AT, information on human breast adipose progenitors is scarce. A better knowledge of these cells is relevant, especially in the context of breast cancer.

Breast cancer is one of the most common cancers in women, with 2.26 million new cases having been diagnosed worldwide in 2020 (https://www.cancer.net/cancer-types/breast-cancer/statistics (accessed on 15 May 2022)). Five major molecular subtypes of invasive breast cancer have been identified, of which approximately two-thirds express estrogen receptors (ERs) that are key drivers of malignant potential [13,14]. AT is a critical energy provider to support the malignant progression of invasive carcinomas, including breast, prostate, and ovarian cancers [15,16]. In addition, obesity is considered as a risk factor for several malignancies, including colon, kidney, pancreatic, breast, and ovarian cancers (WHO, https://gco.iarc.fr/causes/obesity/help (accessed on 15 May 2022)). Moreover, adipocytes secrete adipokines that promote cancer-cell growth or facilitate epithelial–mesenchymal transition [17], and an increase in pro-inflammatory mediators leading to local inflammation is encountered in obese individuals [18]. All of these organs are enveloped by adipose tissue. However, as the lobular structure is destroyed in tumors, adipocytes and APCs come into close contact with cancer cells when malignancies arise [19,20]. 

In breast cancer, changes in AT allow adipocytes to stand in close proximity to tumor cells in the invasive front of the tumor [20,21], whereas they are barely detectable in the tumor core. These adipocytes have been termed “cancer associated-adipocytes” (CAAs). They have smaller lipid droplets and display an increased lipolysis, leading to delipidation. This process may be accompanied by the browning of adipocytes close to the cancer cells, for example, by paracrine secretion of adrenomedullin or other molecules [20,22]. When the lobular structure of the AT is destroyed, APCs also come into contact with cancer cells and acquire the properties of cancer-associated fibroblasts (CAFs). This phenomenon is particularly striking in breast tumors, as ATis abundant in this organ. Of note, obesity increases the rate of conversion of APCs to CAFs [23]. As APCs, CAFs are heterogeneous between different tumor types and even within the same tumor. Based on the markers they express, they can be classified as myofibroblast (Myo) CAFs, inflammatory CAFs, vascular CAFs, cyclic CAFs, or developmental CAFs [24,25], although no clear consensus has been adopted for an explicit classification so far. The differentiation of APCs into MyoCAFs has been reported to be dependent on the presence of a primary cilium and induced by TGFβ [26]. MyoCAFs express α-smooth muscle actin (αSMA). In breast tumors, they are primarily located in close proximity to tumor cells and secrete a collagen-rich matrix that has been shown to facilitate the migration and dissemination of breast cancer cells [27].

Indeed, information on the interactions between breast cancer cells and adipose tissue cells, including APCs, is relevant to provide new insights into the progression of invasive carcinomas. 

Here, we used human breast adipose tissue from women who underwent breast reduction procedures to isolate distinct adipose progenitor cells. After selective digestion with collagenase, we obtained two populations with distinct properties. One population expressed higher levels of *ALPL* and differentiated very efficiently into adipocytes. The second expressed higher levels of *INHBA* and *CD142* and was able to differentiate more efficiently into myofibroblasts in a TGFβ-dependent manner, or to Myo-CAFs in the presence of breast cancer cells, as evidenced by expressions of αSMA and N-cadherin. Furthermore, the latter population expressed the immune checkpoint CD274 (or programmed death-ligand1: PD-L1) in a TNFα-dependent manner, indicating that nonimmune, non-transformed cells with the capacity to sustain malignant progression may play an additional role in immune evasion. We analyzed the interactions of these two populations with ER-positive breast cancer cells. Indeed, the low-adipogenic population facilitated the spread of breast cancer mammospheres compared to the population with higher adipogenic potential. In contrast, the latter population increased the expression of ER-α in T47D cells, an ER-positive human breast cancer cell line. 

Overall, our results provide new information about human breast adipose progenitor cells. They identified the type of APCs that preferentially interact with breast cancer cells. They are relevant and informative to better target a specific cell type in the tumor microenvironment that promotes cancer spread.

## 2. Materials and Methods

### 2.1. Reagents

Unless specified otherwise, all reagents were obtained from Sigma (Saint-Quentin Fallavier, France).

Tissue culture media were obtained from LONZA (Levallois-Perret, France), and fetal calf serum (FCS) was obtained from Dutscher S.A. (Brumath, France).

### 2.2. Patients

Breast APCs were derived from the stroma vascular fraction (SVF) of women who underwent breast reduction procedures. Patients received clear information before surgery, and all of them gave written and signed consent. We worked with adipose tissue that was collected during reconstructive surgery. The harvesting of human adipose tissue was standardized for all the patients and had no functional repercussions for the patient. The breast tissue samples were obtained from the inferoexternal area or from the inferointernal area. 

### 2.3. Cell Culture

The hMADS cells were maintained and differentiated as previously described [28]. They were shown to express a thermogenic signature upon appropriate differentiation conditions [29] and are further referred to in this paper as hMADS–adipocytes. 

Breast APCs were prepared and differentiated as previously described [30]. Adipose tissue was dissected to obtain adipose lobules that were minced. A first centrifugation step allowed for the separation of the fat and lean fractions. Digestion of both fractions was performed with collagenase A (10 mg/mL) in PBS/BSA (2% *w*/*v*) for 30 min, at 37 °C, under gentle stirring. Cells of the stroma vascular fraction (SVF) were collected by centrifugation at 260 g for 5 min and further washed with PBS before plating (see Appendix A for more details).

Adipospheres formation was carried out by incubating 100 cells in 20 µL of complete growth medium as hanging drops for 4 days. They were further grown and differentiated in ultra-low attachment (ULA) plates for 14 days.

Adipocytic differentiation was assessed by Oil-Red-O staining [31]. Counter-staining with Crystal Violet (0.1% *w*/*v*) was performed for 10 min. 

Breast cancer cell lines (MCF7 and T47D) were grown in DMEM (Phenol red free) supplemented with antibiotics, glutamine, and FCS (10% *v*/*v*). Sphere formation was carried out on plates coated with agarose in PBS (1% *w*/*v*). Cells were plated at a density of 10,000 cells/mL in DMEM medium supplemented with B27 nutrient (Invitrogen), 20 ng/mL EGF, and 20 ng/mL FGF2 [32]. Mammospheres were collected by sedimentation and were further used in the experiments. All cell lines were routinely tested for the absence of mycoplasma. 

### 2.4. Gene Expression Analysis 

The TRI-Reagent kit (Euromedex, Soufflweyersheim, France) was used to extract total RNA. Reverse transcription (RT) was performed by using M-MLV reverse transcriptase (Promega, Charbonnieres, France), as recommended by the manufacturer.

All primer sequences are indicated in the Appendix A. Real-time PCR assays were run on an ABI Prism StepOne real-time PCR machine (Applied Biosystems, Courtaboeuf, France). The reference gene used for normalization was *36B4*. Quantification was completed by using the comparative Ct method.

### 2.5. Protein Expression 

Cells were washed with ice-cold PBS, and the whole-cell extracts were prepared as previously described [33]. Briefly, cells were lysed in the cell lysis buffer, and then the lysate was sonicated for 10 s and centrifuged at 12,000× *g* for 10 min.

Unless specified, thirty micrograms of proteins were resolved by SDS–PAGE, under reducing conditions, and further transferred to Immobilon–P membranes (Millipore, Molshiem, France). Antibodies used for detection are listed in the Appendix A. They were used according to the manufacturer’s instructions. We used horseradish peroxidase-conjugated secondary antibody (Promega, Charbonnieres, France) to detect the bound primary antibody. Detection was visualized by using an ECL detection kit (Millipore, Molsheim, France). Chemiluminescence was observed by using a molecular imager Amersham Imager 600 v1.0(GE Healthcare Europe, Freiburg, Germany) imaging system. Band intensity was quantified by using Amersham Imager 600 analysis software or with FIJI [34], as indicated in the figure legends.

### 2.6. Fluorescence-Activated Cell Sorting (FACS)

Cells were dissociated and treated with Fc Block (BD Biosciences, Le Pont de Claix, France) for 10 min at 4 °C. They were then incubated for 1 h at 4 °C with primary antibodies coupled to fluorochrome, as described in the Appendix A. Analysis of the staining was performed by FACS, using a BD LSR Fortessa (BD Biosciences, Le Pont de Claix, France) on 50.000 events.

### 2.7. Immunocytochemistry

Cells were grown on glass coverslips and processed as indicated in the legends. Labeling was performed with antibodies listed in the Appendix A, as previously reported [35]. They were revealed with the appropriate secondary antibody coupled to Alexa Fluor (1:1000). To evaluate the unspecific signal, a control condition without primary antibody and a non-specific antibody was carried out for each antibody. We examined six-to-ten representative fields for each condition. Images acquisition was achieved on a Zeiss Axio Observer microscope (Carl Zeiss S.A.S., Rueil Malmaison, France) with an EC Plan Neofluar 40X (NA 1.3) oil objective, using AxioVision 4.8.2 software (Carl Zeiss Microscopy, White Plains, NY, USA). Analysis was performed by using Fiji [34].

### 2.8. Time Lapse Experiments

Adipose progenitor cells were grown and differentiated on glass coverslips mounted in a specific adaptor for microscopy. Breast mammospheres were added 24 h prior to the beginning of the recording in order for them to adhere to the cells grown on coverslips. The time-lapse experiments were performed on an inverted AxioObserver-Zeiss microscope equipped with a sCMOS ANDOR Neo camera at 37 °C in a CO_2_-controlled atmosphere. The images were acquired every 20 min, with a total imaging time of 63 h. The system was controlled by using MetaMorph software (Molecular Devices, Sunnyvale, CA, USA). Analysis of the area covered by cancer cells was performed by using Fiji [34].

### 2.9. Spinning Disk Experiments 

The spinning-disk experiments were performed on an inverted IX81 Olympus microscope (Olympus, Center Valley, PA, USA). Separation of the emission signals was performed by using a GFP/mCherry filter cube containing a beam splitter 580 and two emission filters, BP 525/50 and LP 600, respectively. The LASER lines were at 405 nm (diode), 488 nm (DPSS), 561 nm (DPSS), and 640 nm (Diode; Andor Technology, Belfast, UK). The MetaMorph software (Molecular Devices, Sunnyvale, CA, USA) controlled the system. A 3D reconstruction was performed by using IMARIS software (Bitplane AG, Zurich, Switzerland).

### 2.10. Statistical Analysis

The results are shown as mean + standard error of the mean (SEM). The number of experiments is indicated. Statistical significance was determined by *t*-tests, using Micrococal Origin 6.0 (Micrococal Software, Northampton, MA, USA) or BiostaTGV (INSERM and Sorbonne University, Paris, France). Probability values <0.05 were considered to be statistically significant and are marked with a single asterisk, double asterisks (<0.01) and triple asterisks (<0.001).

## 3. Results

### 3.1. Heterogeneity of Human Breast APCs

#### 3.1.1. Characteristics of the Two APC Populations

We used adipose tissue obtained from surgical breast-reduction specimens to isolate and characterize cells present in the stromal vascular fraction (SVF). The age and BMI of the patients are summarized in Appendix A. We used samples from patients aged 50 years or younger because the proliferative and differentiation capacity of APCs decreases with age [36,37]. We worked with samples obtained from non-obese patients to characterize the healthy breast APC populations. The AT lobules were minced and further divided into two parts that were digested with collagenase and washed with PBS. Then the cells were plated on Petri dishes. SVF associated with a rather lean fraction is here referred to as Population 1, whereas SVF from the fraction containing more fat cells is here referred to as Population 2 (Appendix A). 

We analyzed the proliferation of the distinct populations. Figure 1A shows that the proliferation of Population 2 was significantly slower than that of Population 1 for Patients 1, 2, and 3. For Patients 4 and 5, no significant difference was measured (Figure 1A,B).

#### 3.1.2. Marker Expression in the Two AP Populations 

To characterize the two populations, we analyzed the expression of different markers (CD73, CD90, CD105, CD142, *INHBA*, and *ALPL*) that were associated with adipose progenitors. 

FACS Analysis:

The FACS analysis of cell surface marker expression revealed that CD73 and CD90 did not discriminate between the two populations, as 100% of the cells were positive for both (Table 1).

The number of CD105-expressing cells was higher in Population 2, ranging from 96 to 100%, depending on the patient, whereas only 65–89% of cells in Population 1 were positive. In addition, the expression level of CD105 was also higher in Population 2 for all patients (Table 1). In contrast, CD142 was preferentially expressed in Population 1. CD142 expression levels in Population 2 were 43% to 74% of those measured in Population 1. These results are in line with those obtained from the RT-QPCR experiments (Figure 1C–G). Thus, Populations 1 and 2 expressed CD105 and CD142 at different levels. 

RT-QPCR Analysis:

Because CD34 expression is a hallmark of native human APCs, we ensured that it was expressed in both populations (Appendix A). We compared its expression to that measured in human AT. Although it was preferentially expressed in Population 2, its expression did not vary significantly from Population 1. We could not detect CD31 and CD45 in either population. 

We then analyzed the expression of markers known to discriminate between high and low adipogenic SVF populations. *INHBA* marks populations with low adipogenic potential and was preferentially expressed in Population 1 in all samples. In contrast, *ALPL* (MSCA1), which characterizes populations with higher adipogenic potential, was preferentially observed in Population 2 for all patients analyzed. The *CD142* expression measured by RT-QPCR was in agreement with the results obtained from the FACS analysis. CD142 expression, which has been described as an inhibitor of adipose differentiation, is inversely correlated with MSCA1 expression. Similar results were observed in all samples analyzed (Figure 1C–G). Thus, based on the expression of markers measured by FACS and RT-QPCR, these two populations can be considered as adipose progenitors but with distinct properties. 

### 3.2. Differentiation Potential of the Two Populations

#### 3.2.1. Adipogenic Differentiation

We further analyzed the adipogenic potential of the two populations. We noted that more lipid-laden cells were present when the differentiation medium was applied to Population 2, whereas few adipocytes were observed in Population 1. After 14 or 21 days of differentiation, Oil-Red-O staining revealed the presence of significantly more cells with lipid droplets in Population 2 compared to Population 1 (Figure 2A–D). Similar results were observed after the Oil-Red-O staining of cells cultured and they were differentiated in three dimensions (Figure 2E,F). These results are in agreement with marker expression, as Population 2, which preferentially expressed MSCA1, was more prone to adipogenic differentiation than Population 1. Indeed, higher levels of PLIN1 were detected in Population 2 cells subjected to differentiation for 14 and 21 days (Figure 2G). Note that, even after 21 days of culture in the differentiation medium, the cells did not express significant levels of uncoupling protein 1 (UCP1) compared to the hMADS–adipocytes used as a positive control [29], thus indicating that adipose differentiation was preferentially directed toward white adipocytes. Thus, using a simple experimental protocol, we were able to isolate two populations of APCs from breast adipose lobules, either endowed with high or low adipogenic potential. 

We have previously shown that an increase in primary cilium size occurs during the commitment of adipose differentiation and is crucial for adipocyte maturation [38]. We measured the size of the primary cilium in both populations. After 3 days on adipogenic medium, the size of the primary cilium was larger in Population 2 than in Population 1 (from 4 to 8 μm for Patient 2) (Figure 2H–J). These observations occurred for all patients tested (Figure 2J), with a 1.2- to 2-fold larger cilium size in Population 2 compared to Population 1. This is consistent with the more adipogenic potential of Population 2. However, at this point, we cannot conclude whether this difference in cilium elongation is a consequence or a cause of the difference in the adipogenic potential of the two populations. 

#### 3.2.2. Myofibroblastic Differentiation

APCs not only differentiate into adipocytes but also into myofibroblasts. We challenged both populations to differentiate into myofibroblasts after TGF-β1 treatment. Figure 3 shows that αSMA and N-cadherin expressions were increased in a dose-dependent manner by TGF-β1 (Figure 3A–C). Indeed, αSMA was not detected in control cells, whereas N-cadherin was detected in both populations. The expression of αSMA was preferentially induced in Population 1, indicating that this population was more prone to acquire a myofibroblastic phenotype. We used the “semi” coculture method of APCs and cancer cells that we developed [26] to evaluate the impact of MCF7 and T47D to trigger the secretome of both cell types to induce the CAF phenotype. This technique allows for the analysis of APC proteins without any interference with cancer cell proteins. Indeed, the incubation of Population 1 cells with cancer cells showed that T47D significantly increased αSMA and N-cadherin expressions in APCs, whereas MCF7 cells had a milder effect. T47D-induced αSMA and N-cadherin expressions were significantly different in Populations 1 and 2 (Figure 3D–F), with a weaker induction observed in Population 2. We measured TGF-RI expression in both populations and observed that it was significantly reduced by 30% onward in Population 2 compared with Population 1 (Figure 4A), consistent with lower myofibroblastic differentiation in Population 2. Therefore, breast tumors are capable of inducing the MyoCAF phenotype of human-breast-derived APCs.

To gain additional insight into the myofibroblastic potential of Population 1, we challenged the cells with different cytokines. We observed that TNFα and IFNγ were not able to promote αSMA expression and myofibroblastic differentiation, whereas IL1β had a reduced effect compared with TGFβ1 (Figure 4B). We did not detect *IL1β* mRNA in the MCF7 or T47D breast tumor cell lines, whereas *TGFβ* and *TNFα* were detected in these cells, highlighting a potential paracrine effect of these two effectors on breast APCs. In the MCF7 cell line, the mRNA of both cytokines was measured at comparable levels, whereas *TNFα* was produced 3-to-8-times more than *TGFβ* in the T47D cells (not shown). Consistent with its role in myofibroblast transition [39,40], we also measured CD274 (PD-L1) expression in both populations. CD274 was hardly, if at all, detectable in control cells, indicating that its expression was dependent on a paracrine stimulus (Figure 4B–D). We found that it was not induced by TGFβ, and that IL1β promoted low expression (Figure 4B), whereas TNFα dose-dependently stimulated preferential expression in Population 1 (Figure 4C–E). Of note, TNF receptor expression did not vary significantly between the two populations (Figure 4A). These observations confirm that Population 1 has multiple characteristics of myofibroblasts. In addition, we analyzed CD274 expression in both populations that were incubated or not with an adipogenic differentiation cocktail. CD274 expression was preferentially detected in Population 1 that had not received the differentiation cocktail, indicating that the culture conditions, but not the complete differentiation cocktail, can induce its expression (Figure 4F). This tightens a preferential expression of CD274 in the population capable of exhibiting the characteristics of Myo-CAFs, whereas the absence of expression in mammary lipid-laden cells or in adipocytes excludes a determining role of this protein in these cells.

### 3.3. Interaction of Cancer Cells with the Two Populations

To explore the interactions between breast cancer cells and the two populations of APCs, we evaluated their impact in promoting the expansion of MCF7 mammospheres. We set up cocultures of the two APC populations with GFP-expressing MCF7 mammospheres. We measured the area covered by MCF7 mammospheres after 24 h (position1) and 87 h (position190) in the presence of Population 1 (Figure 5A–D) or Population 2 (Figure 5E–H). Typical recordings are shown in Appendix A. A statistical analysis of the results obtained from five patients is presented in Figure 5I. We found that the area occupied by mammospheres increased 2.23 + 0.08 (n = 8) times in the presence of cells from Population 1, whereas an increase of 1.88 + 0.09 (n = 10) was measured when mammospheres were incubated in the presence of Population 2 (see Appendix A). Thus, Population 1, which was more prone to differentiate into CAFs, certainly facilitated the expansion of MCF7 breast cancer cells more efficiently compared to Population 2. 

Because MCF7 and T47D cells express estrogen receptors, which play an important role in proliferation, tumorigenesis, and progression of breast cancers, we analyzed the impact of both populations on ER receptor expression by using the same coculture method as the one used in Figure 3. Whereas Population 1 did not significantly alter ER-α and ER-β expressions, we observed that co-incubation of T47D for 4 days with Population 2 significantly increased ER-α expression 1.8-fold, whereas ER-β expression was not significantly altered (Appendix A). This effect was hardly detectable in MCF7 cells, presumably because ER expression levels were higher compared to T47D. Thus, the two populations of APCs have distinct characteristics to promote malignant potential and tumor dissemination. A summary of the results is presented in Table 2.

## 4. Discussion

In this study, we analyzed the properties of human breast APCs that were isolated from patients aged fifty years or younger, because the proliferation and differentiation potential of these cells decreases with age [36,37]. We identified two populations of APCs with distinct differentiation profiles and showed that they interacted differentially with breast cancer cells. 

The heterogeneity of adipose tissue is based on the existence of various types of adipocytes and APCs. Cell-type specificity confers differences in the thermogenic capacity of adipose tissues [41,42], as well as in their lipid metabolism. For example, visceral and subcutaneous ATs show different rates of lipolysis or fatty acid uptake and display a distinct secretion profile for proinflammatory cytokines [41,42,43,44,45]. However, their differences do not lie only in a defined metabolic profile attributed to adipocytes. Adipose depots contain several types of APCs with high or low capacities to differentiate into adipocytes [46]. These cells are also able to produce other specialized cells such as myofibroblasts. In this regard, cells expressing high levels of ALPL (MSCA1) constitute a highly adipogenic population, whereas cells with high levels of CD271 [11] or CD142 have little or no adipogenic potential [47]. Both populations have been encountered in human white fat depots such as visceral or subcutaneous ones [11,46]. However, little is known about human mammary adipose progenitors. Here we show that, as previously reported for other adipose depots, there are distinct populations of progenitors isolated from human mammary lobules. The first population we isolated expressed high levels of INHBA, which has been associated with adipose progenitors in septa [11], and CD142, which marks a population of cells that inhibit adipogenesis in fat depots [47,48]. These markers have been described as preferentially labeling cells with myofibroblastic potential. In contrast, the second population showed preferential expression of ALPL and differentiated more efficiently into adipocytes. This was accompanied by a higher expression of CD105 (endoglin). This marker was detected in hMADS cells obtained from surgical adipose tissue scraps, with higher expression than in adipose progenitors prepared from lipoaspirates [49]. Of note, in human lipoaspirates, CD105 was also associated with human adipose resident microvascular endothelial progenitor cells [50]. Although their expressions are found in several stem cells, including mesenchymal stem cells, CD73 (a plasma membrane protein that catalyzes the conversion of extracellular nucleotides to membrane-permeable nucleosides) and CD90 (THY1) did not discriminate between the two populations.

In mice, the adult breast AT is composed of adipocytes with white adipocyte characteristics and retains a high potential for plasticity. During pregnancy and lactation, they transdifferentiate into adipocytes that are suitable for milk production [8]. Adipose differentiation of human mammary progenitors preferentially generated white adipocytes. Indeed, we detected only a minute amount of UCP1, which marks brown or beige adipocytes, in the lipid-laden cells of Population 2. This observation is in agreement with findings showing that brown fat does not represent a normal constituent of the adult human breast, although human breast hibernomas have been described [8,51]. Furthermore, brown adipocytes were not detected in the healthy human breast, but they were found in close proximity to breast cancer cells of distinct origins: estrogen-positive MCF7 cells, triple-negative MDA-MB-231 cells, and HER2+ tumors [20]. Thus, the presence of brown or beige adipocytes in the human breast represents breast structures rather associated with a pathological context such as cancer or hibernomas [20,52,53].

The primary cilium is an organelle that undergoes essential changes during the early stages of adipose differentiation [38]. We observed that APCs with the highest adipogenic potential had a longer primary cilium for all patients tested and that cilium elongation was preferentially favored in Population 2. This comparison revealed that cilium size was not similar in separate APCs derived from breast tissue of the same patient. Consistent with our previous data highlighting the importance of primary cilium elongation for adipose differentiation [26,38], we can also hypothesize that primary cilium size represents a key determinant in driving cell fate. Population 1 preferentially expressed *INHBA* and *CD142*, two markers that are associated with low adipogenic potential. *INHBA* was expressed more in APCs compared to adipocytes, whereas autocrine and paracrine production of activin A impaired adipocyte differentiation by inhibiting C/EBPβ through the activation of SMAD2/3 in human APCs derived from several AT locations [33,54,55]. In addition, *INHBA* expression was associated with a Myo-CAF profile in pancreatic ductal adenocarcinomas [56]. CD142-positive cells were refractory to adipogenesis while they had the ability to inhibit APCs differentiation in mice and humans [47,48]. In agreement with these reports, our results confirm the low adipogenic and high myofibroblastic potential of this population. Although CAFs consist of a highly heterogeneous cell population, the myofibroblastic phenotype is related to those encountered in close proximity to tumor cells [25,57,58]. TGFβ has been reported to be a major inducer of this phenotype through αSMA expression. When cells were treated with TGFβ, we observed that αSMA was preferentially induced in Population 1 compared to Population 2. A similar expression pattern was followed by N-cadherin, which causes cytoskeletal rearrangement; changes in cell shape, as well as changes in cell–cell and cell–extracellular matrix (ECM) interactions [59]; and reveals a more mesenchymal phenotype. Mammospheres, as classical 2D cultures, were able to produce TGFβ which induced a conversion of Population 1 cells into “activated” CAFs, which are known to play an important pro-invasive role that is similar to that described in ovarian cancers [60,61]. This is in agreement with our observation of preferential tumor spread when mammospheres were incubated in the presence of Population 1 compared to Population 2. In addition, N-cadherin expression was increased in the presence of cancer cells and in a TGFβ-dependent manner. N-cadherin expression activates multiple signaling pathways, including those involved in invasive growth in the mammary gland [62]. Since mammospheres express E-cadherin and the formation of E-cadherin and N-cadherin duplexes promotes the invasive potential of tumor cells [63], these observations are consistent with more efficient migration of cancer cells in the presence of Population 1. Furthermore, while the origin of CAFs remains an open question due to their high migratory potential, our results illustrate the occurrence of a “local reprogramming” of Population 1 into mammary-tumor-supporting cells. In contrast, Population 2 cells might be more consistent with a “non-activated” profile of CAFs, as they express lower levels of basal or stimulated αSMA and N-cadherin to a lesser extent [61]. Another alternative could be that this population of adipose progenitors with high adipogenic potential is not able to acquire a CAF phenotype.

Among the other proinflammatory cytokines, we observed that IL1β was, to a lesser degree compared with TGFβ, capable of inducing a myofibroblastic phenotype. Note that, in contrast to TGFβ, IL1β also promoted the expression of the immune checkpoint molecule CD274. Because IL1β was difficult to detect in our model, we looked for other cytokines with this property. We observed that TNFα promoted a similar induction of CD274, although it did not induce the myofibroblast profile associated with αSMA expression. Neither IL6 nor IFNγ stimulation was able to induce CD274 expression in both populations. Of note, we did not measure CD274 expression in lipid-laden cells obtained after differentiation of human mammary APCs or hMADS cells, in contrast to results from murine adipose differentiation [64].

Our simplistic coculture model does not allow for immune interactions. Thus, we cannot measure possible CD274-dependent protection by the immune system, whereas this can occur in vivo. Immuno-independent effects of CD274 have been reported for cancer cells [65]. We cannot exclude that a similar contribution occurs in APC Population 1. CD274 expression was not induced by TGFβ in Population 1, in contrast to observations reported for normal human lung fibroblasts and in idiopathic pulmonary fibrosis (IPF) models [39]. In the IPF model, fibroblast-intrinsic CD274 is required for the development of pulmonary fibrosis through interaction with SMAD3, resulting in further activation of the Wnt-/βCathenin pathway. These observations underscore the existence of CD274 (PD-L1) effects that are independent of PD-L1/PD1 binding. Furthermore, in intrahepatic cholangiocarcinoma (ICC), myofibroblastic CD274 plays a crucial role in modulating both the tumor microenvironment and tumor growth, independent of immune suppression function [40]. Targeting CD274 in hepatic stellate cells prevents these cells from acquiring a myofibroblastic phenotype and results in a dramatic decrease in ICC growth. We cannot rule out a similar role for CD274 in our breast cancer model. We have not studied the pathways activated by CD274, and they may differ from one cell type to another. However, these results showing that Population 1 can both differentiate into myofibroblasts and express CD274 under appropriate stimuli are consistent with both reports. Together, they indicate that the breast APC population, which is competent to generate CAFs, may also be an important source of CD274 and contribute significantly to cancer immune escape. The existence of similar APC populations with equivalent characteristics remains to be established in other cancers in order to recognize their involvement in immune escape more generally.

## 5. Conclusions

For the first time, using a simple differential digestion method of adipose lobules, we characterized two populations of human breast APCs that showed preferential differentiation into adipocytes or myofibroblasts. Whereas the highly adipogenic population increased ER-α expression in T47D cells, we showed that MCF7 cancer cells were more likely to spread on the low adipogenic population. Indeed, myofibroblasts that remodel the microenvironment played a more detrimental role than lipid-laden cells that are able to provide tumor cells with nutrients and adipocytokines reported to stimulate their growth. At this point, we can hypothesize that Population 1 produces a factor or matrix that promotes mammosphere spread. In contrast, Population 2, which generates more adipocytes, shows “less invasive” activity of MCF7 mammospheres. This observation may result from mechanical or physical constraints that prevent mammosphere propagation or from a factor or matrix secreted by adipocytes that is able to reduce MCF7 tumor cell expansion in the absence of Population 1. Furthermore, the contribution of CAFs to the promotion of an immunosuppressive tumor environment remains to be investigated, as information on the mechanisms involved could be relevant to improve the anticancer efficiency of PD-L1/PD-1 immunotherapy.

## Figures and Tables

**Figure 1 biomedicines-10-01928-f001:**
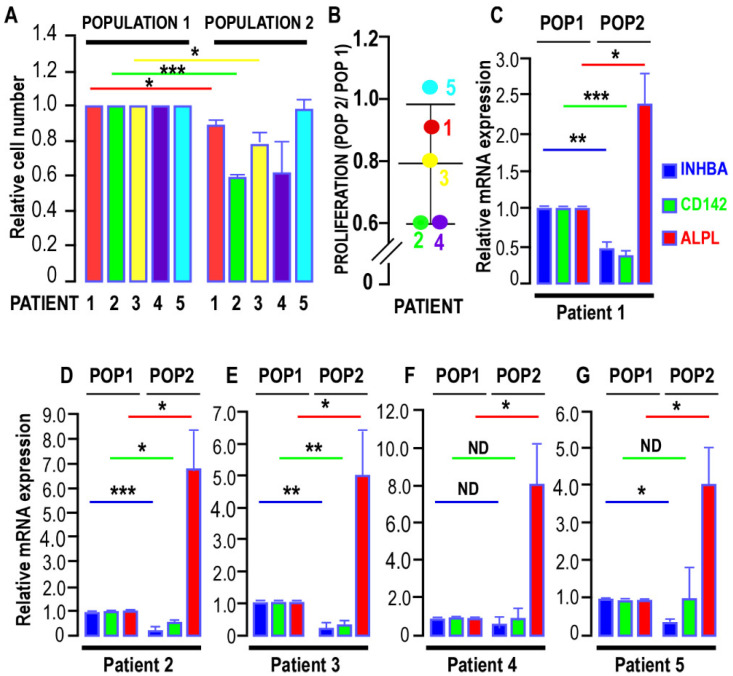
Proliferation and marker expression of the two breast adipose progenitor cell (APC) populations. (**A**,**B**) Proliferation of the two populations of APC isolated from human breast. Cells were seeded in 12-well plates, and proliferation was measured as the number of cells in the wells after 5 days of culture and normalized to the cells present in Population 1. The means ± SEMs were calculated from three independent experiments, with determinations performed in quadruplicate (* *p* < 0.05, *** *p* < 0.001). The ratio of the proliferation rate for Population 2, as compared to Population 1, is presented in (B). (**C**–**G**) Expression of *INHBA, CD142*, and *ALPL* (MSCA1) was assessed by real-time RT-PCR and normalized for the expression of *36B4* mRNA for all the patients. The means ± SEMs were calculated from four (Patients 1, 2, 3, and 4) and five (Patient 5) independent experiments, with determinations performed in duplicate (* *p* < 0.05, ** *p* < 0.01, and *** *p* < 0.001; ND, not different).

**Figure 2 biomedicines-10-01928-f002:**
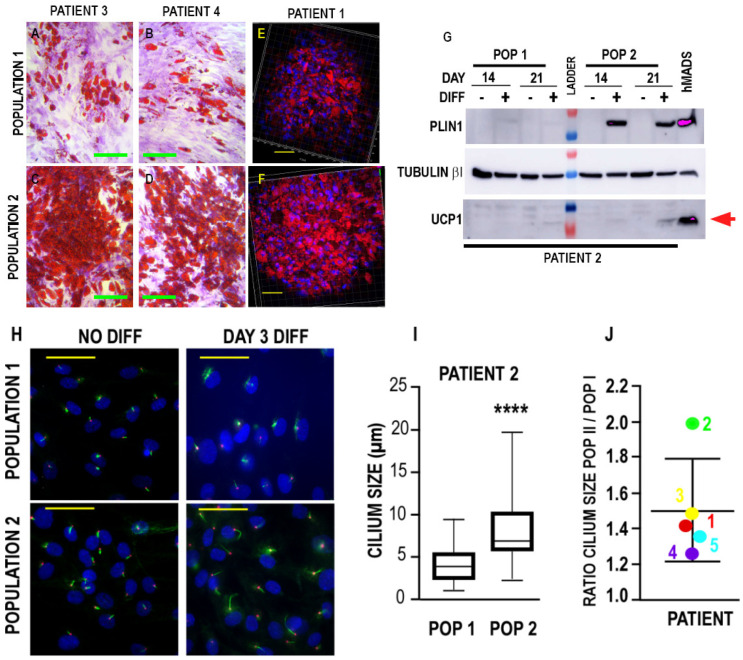
Adipose differentiation of the two breast adipose progenitor cell (APC) populations. (**A**–**D**) Breast APCs were cultured in differentiation medium for 15 days. Differentiation was assessed by Oil-Red-O staining. Crystal Violet was used to counterstain the population that was not differentiated. Magnification ×20; scale bar, 100 µM. (**E**,**F**) APCs were grown and differentiated in 3 dimensions. Oil-Red-O staining revealed that the number of lipid-laden cells is lower in Population 1. DAPI marks the nuclei of the cells. Scale bar, 50 µM. (**G**) Protein expression was measured in breast APCs grown in differentiation medium for 14 or 21 days. Expressions of PLIN1 (upper panel), Tubulin-βI used as a loading control (intermediate panel), and of UCP1 (lower panel) were analyzed by Western blot, using specific antibodies. Representative Western blots are shown. (**H**) Cells were fixed, and acetylated tubulin (green) and pericentrin (red) were revealed by immunocytochemistry; nuclei were stained with Hoechst 33,258 (blue). The yellow scale bar represents 50 µm. (**I**) Quantification of the cilium size of the different populations after 3 days of differentiation. One hundred cilia were measured by using image J for each condition. A box plot of the results for Patient 1 is presented (**** *p* < 0.0001, unpaired *t*-test). (**J**) The ratio of the cilium length of Populations 2 and 1 after 3 days of differentiation is provided for each patient. For each condition, a minimum of 100 cilia were measured.

**Figure 3 biomedicines-10-01928-f003:**
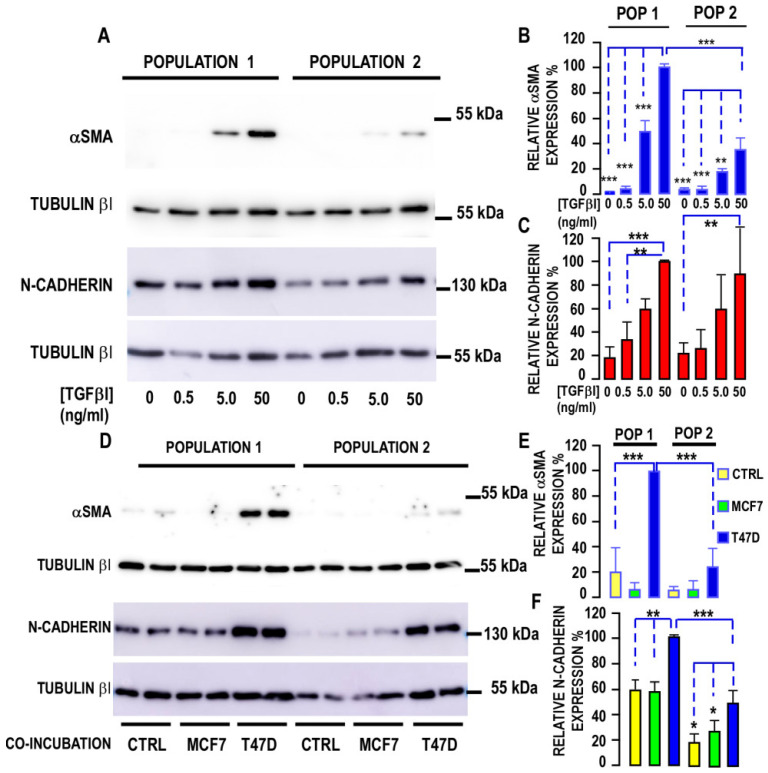
Myofibroblast differentiation of the two breast adipose progenitor cell (APC) populations. (**A**) Protein expression was measured in the two populations of breast APC submitted to increasing concentrations of TGFβ for 3 days. Expressions of αSMA and N-cadherin were measured by SDS–PAGE, using gels with distinct percentage of acrylamide: 10% and 7.5% for αSMA and N-cadherin, respectively. Tubulin-βI was used as a loading control in both cases. Expressions of the two proteins were analyzed by Western blot, using specific antibodies. Representative Western blots are shown. (**B**,**C**) Quantification of the signals. Protein expression was quantified by using FIJI program and compared to the expression of Tubulin-βI. The means ± SEMs were calculated from four independent experiments (** *p* < 0.01, and *** *p* < 0.001). (**D**) Protein expression was measured in the two populations of breast APs co-incubated with MCF7 or T47D cells or without tumor cells (CTRL) for 4 days. Expressions of αSMA and N-cadherin were measured by SDS–PAGE, using gels with distinct percentage of acrylamide: 10% and 7.5% for αSMA and N-cadherin, respectively. Tubulin-βI was used as a loading control in both cases. Expressions of the two proteins were analyzed by Western blot, using specific antibodies. Representative Western blots are shown. (**E**,**F**) Quantification of the signals. Protein expression was quantified by using the FIJI program and compared to the expression of Tubulin-βI. The means ± SEMs were calculated from four independent experiments (* *p* < 0.05, ** *p* < 0.01, and *** *p* < 0.001).

**Figure 4 biomedicines-10-01928-f004:**
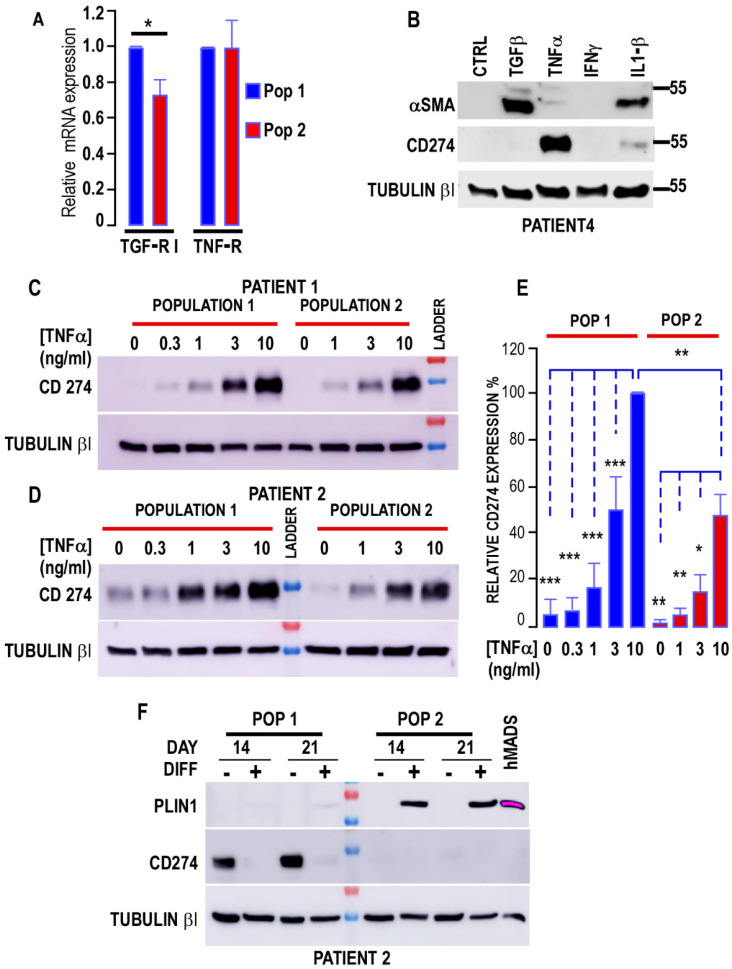
Expression of CD274 in the two breast adipose progenitor cell (APC) populations. (**A**) Expression of *TGF-R1* and *TNF* receptors in cells of the two APC populations. Expression of *TGF-RI* and *TNR* receptors was assessed by real-time RT-PCR and normalized for the expression of *36B4* mRNA. The means ± SEMs were calculated from three independent experiments, with determinations performed in duplicate (* *p* < 0.05). (**B**) The effect of distinct cytokines on myofibroblast differentiation protein expression was measured in Population 1 of breast APCs submitted to 10 ng/mL of TGFβ, TNFα, IFNγ, or IL1β for 3 days. Expressions of αSMA, CD274, and Tubulin-βI (used as a loading control) were measured by SDS–PAGE. Expressions of the proteins were analyzed by Western blot, using specific antibodies. Representative Western blots are shown. (**C**,**D**) TNFα dose-dependently increases CD274 expression. Protein expression was measured in the two populations of breast APCs submitted to increasing concentrations of TNFα for 3 days. Expressions of CD274 and Tubulin-βI was used as a loading control were measured by SDS–PAGE. Expressions of the two proteins were analyzed by Western blot, using specific antibodies. Representative Western blots are shown. (**E**) Quantification of the signals. Protein expression was quantified by using FIJI program and compared to the expression of Tubulin-βI. The means ± SEMs were calculated from five independent experiments (* *p* < 0.05, ** *p* < 0.01, and *** *p* < 0.001). (**F**) Analysis of CD274 expression in differentiated cells. Protein expression was measured in the two populations of breast APCs grown in differentiation medium for 14 or 21 days. Expressions of PLIN1 (upper panel), CD274 (intermediate panel), and Tubulin-βI (used as a loading control) (lower panel) were analyzed by Western blot, using specific antibodies. Representative Western blots are shown.

**Figure 5 biomedicines-10-01928-f005:**
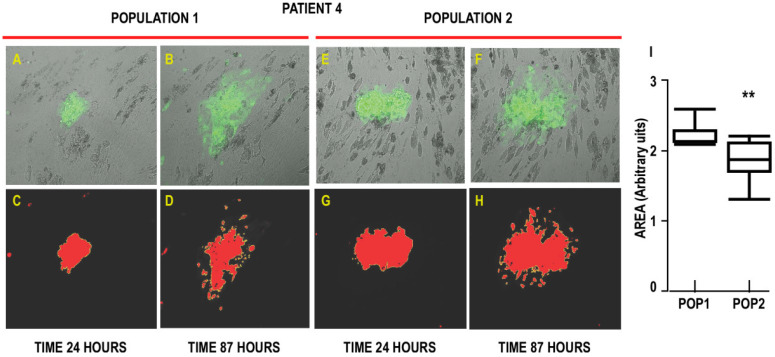
Interactions of the two breast adipose progenitor cell (APC) populations with MCF7 mammospheres. Time-lapse analysis of breast cancer mammospheres spreading on Populations 1 (**A**–**D**) and 2 (**E**–**H**) was performed on cells differentiated for 21 days. They were incubated for 24 h with MCF7 mammospheres expressing GFP prior to recording for the mammospheres to adhere to the cells. Recording lasted for 63 h and was performed every 20 min, using an inverted AxioObserver-Zeiss microscope equipped with a sCMOS ANDOR Neo camera at 37 °C in a CO2-controlled atmosphere (magnification 10×). The area covered by the mammospheres was analyzed by using IMAGE J. The red image delineates the area after defining the threshold level to ignore non-specific signals. The increase in mammosphere size was calculated by the ratio of the areas measured at the beginning and at the end of the experiment. (**I**) Box plots of the data are presented for each patient in Appendix A (** *p* < 0.01).

**Table 1 biomedicines-10-01928-t001:** FACS analysis of breast adipose progenitor cell (APC) markers expression.

Patient	1	1	2	2	3	3	4	4	5	5
Population	**1**	**2**	**1**	**2**	**1**	**2**	**1**	**2**	**1**	**2**
CD73	100	100	100	100	100	100	100	100	100	100
CD90	100	100	100	100	100	100	100	100	100	100
CD105	85.1	96.1	64.5	99.4	88.8	99.5	66.7	97.6	78.6	97.8
CD105 levels	2474	3219 (135%)	712	7426 (1042%)	1252	1918 (153%)	571	1571 (275%)	794	1380 (173%)
CD142	79	53.3	90	90	98	91	83.9	79.9	90	83
CD142 levels	911	604 (66%)	12,400	5400 (43.5%)	3979	2477 (62%)	990	736 (74%)	3048	1401 (46%)

Note: The values indicated in blue represent the percentage of positive cells for each population type. For CD105 and CD142 levels, the mean fluorescence intensity (black) of the signal is given. The values in parentheses represent the expression level of Population 2 as compared to Population 1. The red and pink colors were used to better differentiate the patient number and the population number.

**Table 2 biomedicines-10-01928-t002:** Summary of the properties of the two breast APC populations.

	Population 1	Population 2
**Proliferation**	**++**	**+**
**Markers**		
CD105	Lower	higher
CD142	High	low
*ALPL*	Low	high
*INHBA*	High	low
**Adipose Differentiation**	**+**	**+++**
**Myofibroblast Differentiation**	**+++**	**+**
**CAF Differentiation**	**+++**	**+**
**PD-L1 Expression**	**++**	**+/−**
**Breast Cancer Migration**	**++**	**+**

Note: +/− means observed for some, but not all patients, + means observed for all patients, ++ means highly observed, +++ means very highly observed.

## Data Availability

Data are contained within the article or Appendix A.

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
