# Peer review of "Identification of Human Breast Adipose Tissue Progenitors Displaying Distinct Differentiation Potentials and Interactions with Cancer Cells"

_biomedicines, 2022, doi:10.3390/biomedicines10081928_

Round 1
Reviewer 1 Report
The study by Peraldi et al. identifies two populations of human breast adipose tissue progenitor cells that display distinct differentiation potentials and interactions with breast cancer cells. The two adipose progenitor cell types were isolated from human breast adipose tissue lobules by differential enzymatic dissociation and were shown to selectively express different sets of marker proteins. The cell population that preferentially expresses ALPL(MSCA1) displays a high adipogenic potential and differentiates towards white adipocytes. The cell population expressing higher levels of INHBA and CD142 acquired myofibroblast features upon Transforming Growth Factor-beta (TGF-beta) treatment and a myo-cancer associated fibroblast profile in the presence of breast cancer cells. This cell population expressed the CD274 immune checkpoint in a tumor necrosis factor-alpha dependent-manner, and the cells facilitated the spreading of MCF-7 cell breast cancer mammospheres compared to the adipogenic cell population.
The investigators conclude that two distinct populations of human breast adipose tissue progenitor cells can be isolated that can interact preferentially with breast cancer cells. The results provide new insights on the existence and potential functions of human breast adipose progenitor cells, which advances the known information in this field. The investigators’ detailed analyses of the adipose progenitor cell types comprehensively defines the two cell populations that express distinct cell markers and display differences in how they interact with breast cancer cell lines. The presented results are generally straightforward to interpret, and the information will be useful to other investigators. The key strength of the study is the rigorous characterization of the two distinct populations of human breast adipose tissue progenitor cells. However, less convincing are the experiments employing either two breast cancer cell lines (T47D and MCF7) or only the MCF7 cells because only a limited number of breast cancer cell lines were used to draw conclusions. These studies require more in-depth analyses to fully establish critical differences in the adipose progenitor-breast cancer cell interactions. The following are specific comments and suggestions for the investigators to consider:
1) Establishing the effects of each of the adipose progenitor cell populations on breast cancer cell lines is a generally useful and important approach. However, for the experiments involving TGF-beta treatment (Figure 3) only two breast cancer cell lines (T47D and MCF7) were used; whereas, for the mammosphere spreading experiments (Figure 5) only the MCF7 cells were used. From this very limited set of breast cancer cells, the investigators attempt to conclude that the adipose progenitor cells display differences in how they interact with breast cancer cells. In addition, the investigators seem to ignore the breast cancer phenotypes (such as estrogen responsiveness) and genetic backgrounds of the cell lines used in their study, which is important to the breast cancer field. The investigators need to significantly expand both Figure 3 and Figure 5 to include a broader spectrum of human breast cancer phenotypes that better represent the range of clinically relevant breast cancers. For example, the investigators need to add estrogen receptor negative cells, triple positive, and triple negative breast cancer cell lines to their study as well as breast cancer cells expressing wild type versus mutant p53. This revision will help to establish whether the differential effects of the adipose progenitor cells may be altered depending on the breast cancer phenotype being tested.
2) The adipose cell progenitor cells could potentially be regulating the expression of estrogen receptors and/or HER2 in individual breast cancer cells. The investigators should characterize the receptor profiles in the breast cancer cells used in their experiments before and after interactions with each of the adipose progenitor cells.
3) As a minor but important revision, it would be helpful to the reader to add a table that clearly and succinctly summarizes the key properties of each of the two human breast adipose tissue progenitor cells identified in the study.
Author Response
We thank reviewer 1 for his interest in our work.
POINT 1/
1) Establishing the effects of each of the adipose progenitor cell populations on breast cancer cell lines is a generally useful and important approach. However, for the experiments involving TGF-beta treatment (Figure 3) only two breast cancer cell lines (T47D and MCF7) were used; whereas, for the mammosphere spreading experiments (Figure 5) only the MCF7 cells were used. From this very limited set of breast cancer cells, the investigators attempt to conclude that the adipose progenitor cells display differences in how they interact with breast cancer cells. In addition, the investigators seem to ignore the breast cancer phenotypes (such as estrogen responsiveness) and genetic backgrounds of the cell lines used in their study, which is important to the breast cancer field. The investigators need to significantly expand both Figure 3 and Figure 5 to include a broader spectrum of human breast cancer phenotypes that better represent the range of clinically relevant breast cancers. For example, the investigators need to add estrogen receptor negative cells, triple positive, and triple negative breast cancer cell lines to their study as well as breast cancer cells expressing wild type versus mutant p53. This revision will help to establish whether the differential effects of the adipose progenitor cells may be altered depending on the breast cancer phenotype being tested.
We thank reviewer 1 for this very important point. The main objective of our manuscript was to characterize two different populations of adipose progenitor cells (APC) in breast adipose tissue, which were easily isolated and showed different properties. This was observed through their ability to differentiate preferentially into adipocytes or myofibroblasts or cancer associated fibroblasts (CAFs). Now, we also show that ER receptor expression in ER-positive T47D cells varies differently upon contact with the two populations of APCs (see below), highlighting an increased difference between the two populations of APCs. Although further work on the effect of different types of breast cancer on the two populations of APCs would be very interesting, it is beyond the scope of this manuscript. Indeed, the major interest of our laboratory is to decipher the properties of the cells composing the adipose tissue in a physiological and pathological context. We have previously established 3D and co-culture models of APCs or adipocytes with breast cancer cell lines that are currently available and used in our laboratory to analyze the interactions between adipose tissue and breast cancer cells (Paré et al 2020 PMID 32819314, Peraldi et al 2020, PMID 33049976). Although widely used by the scientific community, we are well aware that these cell lines do not recapitulate all the characteristics of breast tumor specimens that we unfortunately cannot obtain.
We took into account the presence of estrogen receptors on the MCF7 and T47D cell lines and to avoid any interference with estrogen reactivity, our experiments were performed with media without phenol red.
To perform the time-lapse analysis of the interactions between breast cancer cells and cells of the two APCs fractions, we used MCF7 mammospheres because the shape of the obtained mammospheres is round and well delimited which is not always the case with T47D and even MDA-MB-231 cells. This point is crucial for a correct analysis and represents a limitation for further studies because it is not possible to perform this type of analysis if the mammospheres are not compact enough or if they undergo a fragmentation process.
To meet the referee's requirement, we have minimized the scope of our results by always referring to the cell lines used in this study.
POINT2/
2) The adipose cell progenitor cells could potentially be regulating the expression of estrogen receptors and/or HER2 in individual breast cancer cells. The investigators should characterize the receptor profiles in the breast cancer cells used in their experiments before and after interactions with each of the adipose progenitor cells.
To address the reviewer's comments, we performed additional experiments that are now presented in Supplementary Figure 3 of the new version of our manuscript. Briefly, we measured estrogen receptor expression on MCF7 and T47D cells incubated without adipose progenitor cells or with population 1 or population 2 by western blot using the co-culture protocol that was used to analyze myofibroblast differentiation (see Figure 3). We observed that ERβ receptor expression is not significantly altered in the tested breast cancer cell lines. However, ERα is increased twofold onwards in T47D cells incubated with the adipocyte progenitor population 2. These data provide new information regarding the differences between AP populations.
POINT3/
3) As a minor but important revision, it would be helpful to the reader to add a table that clearly and succinctly summarizes the key properties of each of the two human breast adipose tissue progenitor cells identified in the study.
The requested table (Table 2) has been added before the discussion section.
Reviewer 2 Report
This is an interesting manuscript with some tech limits (a single-cell RNA analysis would have been more informative), but data are clear and well described. I have only a few suggestions:
Please comment the age range of healthy subjects involved in the study. Does the literature suggest significant variations in breast APC frequency and activity in relationship with age? Any relationship with BMI?
Were all APCs CD34+CD45-CD31-? A flow cytometry dot plot figure would be useful.
Data about the interaction of MCF-7 mammospheres with APCs are interesting. However, I would comment more on the fact that both population 1 and population 2 increase mammosphere formation, with population 1 showing only a modest advantage.
Author Response
Comments and Suggestions for Authors
This is an interesting manuscript with some tech limits (a single-cell RNA analysis would have been more informative), but data are clear and well described. I have only a few suggestions:
POINT 1/ Please comment the age range of healthy subjects involved in the study. Does the literature suggest significant variations in breast APC frequency and activity in relationship with age? Any relationship with BMI?
We thank reviewer 2 for his interest and his valuable comments.
Aging decreases the proliferation and differentiation capacity of APCs (Schipper et al. 2008 PMID 18434829; Kirkland et al. 2020 PMID 12175476), resulting in poor adipogenesis in the elderly. We have also experienced this in previous work using APCs obtained from elderly patients. For this reason, we chose to work with APCs from rather young patients. We did not notice any major differences between the APCs of the younger and older patients analyzed here.
We have added the sentence “We used samples from patients aged 50 years or younger, because the proliferative and differentiation capacity of APCs decreases with age [36, 37].” page 5 and the corresponding references to explain our choice. (PMID 12175476, and 18434829)
When fat mass increases in obese individuals, the overall number of APCs is increased and associated with a reduced capacity to differentiate into fat cells (Tchkonia et al. 2010, PMID 20701600). Here, the BMI of the patients was given to indicate that we are not working with samples from obese patients. We chose to work with adipose tissue derived from non-obese patients to assess the characteristics of the "healthy" breast-derived APCs. Of course, knowing that obesity is considered a risk factor for cancer development, analysis of APCs from obese patients would have been interesting. But breast cancer does not only develop in obese patients. Another aspect is that our laboratory does basic research but not clinical research, and we have access to a limited number of patient samples. Comparing adipose progenitors from non-obese and obese patients requires a much larger number of samples to perform a proper statistical analysis, and at this point we do not have the ability to undertake such a study.
POINT2/
Were all APCs CD34+CD45-CD31-? A flow cytometry dot plot figure would be useful.
We did not perform FACS analysis on native human breast APCs, as we focused on culture expansion. Because CD34 expression was reported to decrease during in vitro expansion of adipose progenitors (Scherberich et al. 2013, A familiar stranger: CD34 expression and putative functions in SVF cells of adipose tissue, PMID: 23362435), we preferred to use markers whose expressions do not vary depending on the cell culture. We analysed marker expressions by RT-QPCR on both fractions as we work with primary cultures. We used human adipose tissue cDNA as a positive control. CD34 expression was detected at very low levels in both APC populations. The results are shown in Supplementary Figure 2.
We did not detect CD31 and CD45 mRNAs in either population. Note that the medium used does not allow for the development of immune or endothelial cells. The residual cells are eliminated as soon as the cells are dissociated. We also did not detect the NGF receptor (CD271) whose expression is lost in culture.
POINT3
Data about the interaction of MCF-7 mammospheres with APCs are interesting. However, I would comment more on the fact that both population 1 and population 2 increase mammosphere formation, with population 1 showing only a modest advantage.
Mammospheres were formed independently of the presence of adipose progenitors. Mammosphere formation requires serum-free culture medium and an incubation on agarose-coated dishes. Both of these requirements are not at all suitable for growing or even maintaining APCs. Thus, the culture conditions that have been optimized to obtain mammospheres do not allow us to evaluate the impact of APCs on mammosphere formation. We simply analysed how mammospheres spread over the two populations of adipose progenitors. At this stage, we can make two hypotheses: either population 2, which produces more adipocytes, has an "anti-tumor" activity and blocks the spread of mammospheres due to physical constraints, or because the lipid-laden cells secrete a factor (or a special matrix) that reduces the expansion of tumor cells, independently of the immune system. In a parallel manner, the second hypothesis is that population 1 may produce a factor or matrix that promotes expansion. These options are now proposed and discussed on pages 14-15.
Reviewer 3 Report
My concern is that results from only five patients are likely to be somewhat incidental or anecdotal. I think their study is interesting and has a potential for clinical implication.
Author Response
My concern is that results from only five patients are likely to be somewhat incidental or anecdotal. I think their study is interesting and has a potential for clinical implication.
We thank reviewer 3 for his interest in our study. We would like to emphasize that we were able to define two distinct APC profiles, each of which was similar in the five patients. Characterization led to homogeneous profiles despite individual differences in the magnitude of the responses.
Round 2
Reviewer 1 Report
The revisions incorporated into the manuscript are appropriate and strengthen the overall study.